# Factors That Affect Symptoms of Injection Site Infection among Japanese Patients Who Self-Inject Insulin for Diabetes

**DOI:** 10.3390/healthcare9040402

**Published:** 2021-04-01

**Authors:** Yuko Yoshida, Masuko Sumikawa, Hiroyuki Sugimori, Rika Yano

**Affiliations:** 1Department of Fundamental Nursing, Faculty of Health Sciences, Hokkaido University, Sapporo 060-0808, Japan; r-yano@med.hokudai.ac.jp; 2Department of Nursing, School of Health Sciences, Sapporo Medical University, Sapporo 060-8556, Japan; masuko0811@sapmed.ac.jp; 3Department of Biomedical Science and Engineering, Faculty of Health Sciences, Hokkaido University, Sapporo 060-0808, Japan; sugimori@hs.hokudai.ac.jp

**Keywords:** disinfection, diabetes mellitus, infections, skin, subcutaneous injections

## Abstract

In Japan, skin disinfection is typically considered necessary before an insulin injection to prevent infection at the injection site. This cross-sectional study evaluated factors that influenced symptoms of injection site infection among 238 Japanese patients who self-injected insulin for diabetes between October 2015 and January 2016. A structured questionnaire was used to collect data regarding skin disinfection practices, infection symptoms at the injection site, frequency of injections, environment at the time of injection, and hygiene habits. The majority of patients (83.2%) performed skin disinfection before the self-injection. Logistic regression analysis revealed that infection at the injection site was positively associated with skin disinfection before injection, age, and performing injections outside home. It was speculated that omitting skin disinfection before administering subcutaneous insulin injection was not the factor that affected the symptoms of injection site infection. The greatest contributor to infection symptoms was injections performed outside the home. Future studies focusing on the environment, in which patients administer insulin injections, to assess its influence on symptoms of injection site infections are warranted.

## 1. Introduction

Skin disinfection before injection is a common procedure that is performed based on the idea that the needle disrupts the skin and creates a risk of infection from bacteria entering the body. Thus, antiseptic techniques are recommended to disinfect the skin before the insertion of peripheral venous catheters and intravascular catheters, which are commonly used devices [1]. However, only washing the skin with soap and running water is recommended before subcutaneous injections based on the World Health Organization’s Best Practices for Injections and Related Procedures Toolkit [2], which also indicates that skin wipes with alcohol or other disinfectants are not essential. The Forum for Injection Technique UK [3] and Australia’s Department of Health [4] have also reported that disinfection is not necessary before a subcutaneous injection. A randomized controlled trial conducted by Wong et al. [5] using intramuscular and subcutaneous injections revealed that omission of skin disinfection did not induce infectious symptoms. Moreover, swabbing the clean skin of a patient is unnecessary before administering an injection, although skin that is visibly soiled or dirty must be washed [6]. Disinfection is also usually not required when injections are administered in non-institutional settings, such as homes, workplaces, or restaurants [7].

Subcutaneous injections are commonly performed for patients with diabetes, and poor diabetes control might be the cause or the consequence of infection in this setting [8]. For example, hyperglycemia may be related to abnormal microcirculation, peripheral vascular disease, skin aging, impaired leukocyte function, and diabetes-related immunosuppression [9]. Thus, as patients with diabetes are susceptible to infection, it is generally considered important to perform skin disinfection before subcutaneous injections for these patients. Nevertheless, some studies of patients who self-inject insulin for diabetes have suggested that the risk of infection is not increased in the absence of skin preparation. Using a pre-test/post-test design and a 3–5-month study period, Koivisto and Felig [10] found no cases of local or systemic infection among 13 patients who received >1700 insulin self-injections with and then without skin preparation. Similarly, McCarthy et al. [11] studied 50 patients who received 1800 insulin self-injections during a crossover trial that involved skin preparation with alcohol or tap water, or no skin preparation, and reported that none of the patients experienced injection site complications.

Among the patients with diabetes who routinely self-administer subcutaneous insulin injections, the rates of pre-injection skin disinfection are 16% in Spain [12] and 30.0% in Italy [13]. Yoshida and Rano [14], based on a review of the literature in Japan, noted that Japanese patients still routinely perform skin disinfection before subcutaneous insulin injections; there is a gap between Japan and other countries in the implementation rate of skin disinfection before subcutaneous injection. However, little is known about the actual situation of patients with skin disinfection before subcutaneous injection and the relationship between the presence or absence of disinfection and infection in Japan. Therefore, we aimed to clarify the practice of skin disinfection among Japanese patients who self-inject insulin and determined whether skin disinfection was associated with a lower likelihood of injection site infection, to critically evaluate the necessity of pre-injection skin preparation. This information will help determine the actual practice of skin disinfection among Japanese patients who self-inject insulin for diabetes, as well as the factors that might influence infection symptoms at the injection site.

## 2. Materials and Methods

### 2.1. Study Design

This observational cross-sectional study evaluated Japanese patients who self-injected insulin for diabetes to determine whether the omission of pre-injection skin disinfection was associated with infection symptoms at the injection site.

### 2.2. Participants

Convenience sampling was used to select participants from three hospitals in Sapporo between October 2015 and January 2016. Patients were considered eligible if they had been diagnosed with diabetes and were self-injecting insulin to manage their diabetes. Patients were excluded if they were being treated via a continuous subcutaneous insulin infusion, if they had dementia or mental disorders, or if they were <20 years old. Patients were ultimately included if they visited a participating hospital during the study period, consented to participate, and read and completed the questionnaire.

Patients were selected by physicians, who then informed the researchers about the time they (researchers) could contact the patients at the hospitals, which was before or after their scheduled appointment. The researchers provided information regarding the study, including its purpose and procedures, and answered any questions before the patients were enrolled. The questionnaire required approximately 5–10 min to complete, and a researcher was present to assist the patients if they experienced problems completing the questionnaire. Initially, 307 potential participants were identified; among them, 69 patients were excluded because data regarding glycated hemoglobin (HbA1c) concentration and infection symptoms at the injection site were missing in their questionnaire. The sample of eligible patients was considered adequate for the analysis, as it exceeded the required sample size of 153 patients, which was calculated using G*Power3.1 (G*Power 3.1, Universität Kiel, Kiel, Germany). This software is widely recognized for calculating the required sample size of epidemiological studies. In this study, the sample size was calculated based on a power of 0.90 (Power = 1 − β) while the probability of type I error (α) was set at 0.05 for medium effect size [15]. The software procedure was according to previous studies [16].

### 2.3. Questionnaire Development

A self-administered questionnaire was developed with reference to previous studies [14,17] and several discussions among researchers. A pilot test was conducted on two non-medical people. To evaluate the content validity, a discussion with two Certified Diabetes Educators of Japan and pilot tests were conducted several times to determine whether the questionnaire covered the experienced signs of skin infections and items that may be related to them (i.e., including whether skin disinfection before subcutaneous injection was used and whether the researcher’s intent of the questions was conveyed).

#### 2.3.1. Clinical and Demographic Characteristics (7 Items)

Data were also collected regarding patient age, sex, employment status, duration of diabetes, type of diabetes, most recent HbA1c concentration, and number of injections per day, as a high number of injections would presumably increase the risk of infection.

#### 2.3.2. Injection Site Disinfection Practice (2 Items)

Participants were asked to identify their skin disinfection practices using a 5-point Likert scale (scores ranging from 1 (rarely) to 5 (always)). Responses were used to categorize patients into a disinfection group (score 4 (usually) or 5 (always)) and a non-disinfection group (score 1 (rarely), 2 (infrequently), and 3 (sometimes)). The patients’ perceptions of skin disinfection were also evaluated using a 4-point Likert scale (score 1 (absolutely necessary), 2 (somewhat necessary), 3 (not very necessary), and 4 (not necessary)). The responses were used to categorize the patients into groups that considered skin disinfection necessary (score 1 (absolutely necessary) and 2 (somewhat necessary)) or unnecessary (score 3 (not very necessary) and 4 (absolutely unnecessary)).

#### 2.3.3. Infection Experience (2 Items)

Participants were asked if they ever experienced an infection at the injection site. In addition, multiple questions were used to collect data regarding symptoms of infection at the injection site for those who reported having been infected. These questions addressed heat, redness, swelling, pain, and discharge [18], which are the main symptoms of skin and soft tissue infections.

#### 2.3.4. Safety Issues (4 Items)

Hygiene habits (bathing or showering frequency) were evaluated because the World Health Organization guidelines [2] indicate that skin disinfection is unnecessary before subcutaneous injection if the patient has adequate hygiene. Participants were asked how often they take a shower or bath in a week. There is some evidence that needle re-use may be associated with skin infection [7,19]. Thus, the questionnaire also asked the patients to report their frequency of needle re-use. The questionnaire also included an item regarding whether the patients performed injections outside their home, to consider the influence of a poorly prepared environment. If a patient reported performing injections outside their home and for those who answered that they had received injections outside their homes, specific locations were given by multiple-answer questions.

### 2.4. Statistical Analysis

All data were analyzed using the SPSS software (version 22.0; IBM Corp., Armonk, NY, USA). Clinicodemographic characteristics, disinfection habits, perception of disinfection importance, and infection symptoms at the injection site were reported as mean ± standard deviation or number (percentage), as appropriate. Differences were considered statistically significant at *p*-values of < 0.05. The chi-squared test and Fisher’s exact test were used to compare the groups that did and did not perform disinfection, and the groups that did and did not experience symptoms of infection, in terms of infection symptoms, perception of disinfection importance, needle re-use, injections performed outside the home, sex, employment status, type of diabetes, and hygiene habits. The t-test was used to compare the groups in terms of age, HbA1c concentration, duration of diabetes, and the number of injections. Adjusted logistic regression models (forward stepwise selection based on likelihood ratio and forced entry) were used to identify factors that were independently associated with infection symptoms at the injection site.

### 2.5. Ethics

The participants voluntarily participated in the study. This research project was conducted in accordance with the principles of the Helsinki Declaration. The study protocol was approved by the Ethics Committee of the Graduate School of Health Sciences, Hokkaido University (15–65) the Clinical Research Ethics Review of the Sapporo Medical University Hospital (272–5007) and has been approved in the other participating hospitals (6 November 2015, 25 November 2015). All participants received assurances regarding confidentiality and anonymity and provided written informed consent before enrollment.

## 3. Results

The patients’ characteristics are shown in Table 1. The 238 participants included 125 men (52.5%) and 113 women (47.5%), with a mean age of 63.88 ± 12.92 years. The mean duration of diabetes was 17.48 ± 11.29 years and the mean HbA1c concentration was 7.84 ± 1.42%. The mean number of injections per day was 2.69 ± 1.27, and 166 participants (69.7%) reported performing injections outside their homes. The diabetes was classified as type 1 diabetes for 52 patients (21.8%), type 2 diabetes for 140 patients (58.8%), and unknown for 46 patients (19.3%). Bathing or showering were performed daily by 91 patients (38.2%), every 2–3 days by 129 patients (54.2%), and every ≥4 days by 18 patients (7.6%). Forty patients (16.8%) reported omitting skin disinfection before the subcutaneous injections, and significant inter-group differences were observed in terms of perceived disinfection importance (*p* = 0.001), and needle re-use (*p* = 0.012) (Table 1). As for the locations of injection administration outside home, they were most commonly performed in a washroom (47.0%), restaurant (39.2%), or car (32.5%). Six patients provided written responses that indicated they attempted to avoid being seen by others while performing the self-injection. The most common injection symptoms at the injection site included redness (29.8%), pain (21.2%), bruising (12.5%), swelling (10.6%), and itchiness (10.6%).

A comparison of the groups with and without infection symptoms at the injection site (Table 2) revealed significant inter-group differences in terms of sex (*p* = 0.031), age (*p* = 0.001), number of injections/day (*p* = 0.001), injections performed outside the home (*p* = 0.001), type of diabetes (*p* = 0.017), needle re-use (*p* = 0.028), and hygiene habits (*p* = 0.028). Table 3 shows the results of the multivariable logistic regression analyses of factors that were associated with infection symptoms at the injection site. Potentially contributing factors were injections performed outside the home, skin disinfection, age, sex, type of diabetes, number of injections per day, and hygiene habits. Nominal variables were processed as dummy variables and the absence of multicollinearity was confirmed to ensure that the independent variables were not correlated. Infection symptoms at the injection site were independently associated with disinfection of the injection site, age, and injections performed outside the home. As a result of forced entry, there were no confounding factors presented.

## 4. Discussion

This study aimed to clarify whether Japanese patients with diabetes performed skin disinfection before insulin self-injections, as well as evaluate other factors that were associated with infection at the injection site to critically evaluate the necessity of pre-injection skin preparation. A majority of the patients (83.2%) performed skin disinfection before the self-injections, which is substantially higher than the rates reported in other countries, 16% in Spain [12] and 30.0% in Italy [13]. Furthermore, 88.7% of the patients perceived that skin disinfection was necessary, which likely explains the large proportion of patients who performed this step. A comparison of patients who did and did not perform skin disinfection revealed significant differences in terms of HbA1c concentration, perception that skin disinfection is necessary, employment status, and needle re-use. The group that did not perform skin disinfection had higher HbA1c concentrations and were more likely to re-use needles. In this context, Japanese nurses have been educated that skin disinfection is necessary before all subcutaneous injections [20], which suggest that patients receive similar instructions from their nurses. The omission of that step, which was instructed to be necessary, was considered to be a possible omission of other procedures that were taught to be necessary. Thus, patients with poor injection technique may have omitted skin disinfection. Employed patients were also less likely to perform skin disinfection compared to unemployed patients. According to the answers for the question regarding the locations outside home, six patients mentioned that they attempted to avoid being seen by others while performing self-injections. When these factors are considered together, we believe that it would be difficult for employed patients to be alone at a point in time during work hours, and hence, may have chosen to skip the skin disinfection process to administer injections quickly.

Logistic regression analysis revealed that infection at the injection site was positively corelated with skin disinfection before injection, age, and performing injections outside home. O’Neill et al. [21] have also reported that skin disinfection was associated with significantly more signs of skin infection, which might be related to patients who perform routine disinfection being more likely to remember potential symptoms of infection. Disinfection is recommended prior to injection administration to prevent infection [22]. However, in this study, we found it was clarified that the administration of skin disinfection was related to infection symptoms at the injection site, and that there was no relationship between omission of disinfection and signs of infection. Therefore, we speculate that omission of disinfection during subcutaneous insulin injections does not cause infection.

Skin aging can increase the risk of bruising and infection [23], which prompted us to consider the potential contribution of age. However, age was not considered a significant factor, as the odds ratio was approximately 1. Relative to age, injections performed outside the home had a greater effect on infection symptoms at the injection site. It is possible that skin conditions might be incorrectly interpreted by the patient as an infection symptom, such as bruising that is related to a poor puncture technique [7]. In this context, patients may be motivated to perform injections as rapidly as possible when in public settings, which may reduce their likelihood of adhering to safe injection practices [24,25]. Although those studies considered people who inject non-therapeutic drugs, it is possible that patients who self-inject insulin might have similar experiences when performing injections outside their homes. Approximately one-half of our patients who performed injections outside their homes performed the injection in a washroom, which agrees with the proportion (53.8%) that was reported by Strauss et al. [26]. Thus, washrooms appear to be a popular out-of-home location for injections, which may contribute to a poor environment or poor injection technique that ultimately leads to local skin problems. However, further study about the association of the detailed environment, administration conditions, and administration methods with signs of infection at the injection site are warranted. These studies may be associated with improved quality of life among patients who perform insulin self-injections for diabetes.

### Limitations

The limitations of the current study are subjectiveness, small sample size, and generalizability. Therefore, it is difficult to draw definitive conclusions; further research including a larger sample is necessary to reduce uncertainty and conclude on the effects of omitting skin disinfection before subcutaneous injection. Identification of signs of infection was not determined by experts, and the data were self-reported by the patient, not from medical records. Recall bias is a significant issue in studies that have self-reporting. The self-administered questionnaire could not differentiate between true infection symptoms and other local events, such as bruising that was related to a poor puncture technique and allergic reactions. Thus, it is possible that our patients reported events that were local allergic reactions, rather than symptoms of skin infection, although it is important to note that the omission of skin disinfection was not associated with skin infection. While we identified a relationship between infection symptoms and injections performed outside the home, there might have been confounding variables not explored in the current study that likely contributed to interactions between injection site infection experience and patients’ practices. However, as this was a cross-sectional study, causality was not clear and there might have been a possibility of reverse causality. Further studies using longitudinal studies are needed to clarify the causal relationship.

## 5. Conclusions

We found that infection at the injection site was positively correlated with skin disinfection before injection, age, and performing injections outside home, and the greatest contributor to infection symptoms was injections performed outside the home in Japanese patients who self-inject insulin for diabetes. Omitting skin disinfection before the insulin injection was not the factor that affects symptoms of injection site infection.

## Figures and Tables

**Table 1 healthcare-09-00402-t001:** Univariate analyses of clinicodemographic characteristics according to disinfection habit.

Variable	All (*n* = 238) (%)	Disinfection (*n* = 198) (%)	No Disinfection (*n* = 40) (%)	*p*-Value
Sex				
Male	125 (52.5)	100 (50.5)	25 (62.5)	0.224
Female	113 (47.5)	98 (49.5)	15 (34.3)	
Age, years	63.88 ± 12.92	63.77 ± 12.89	60.00 ± 11.62	0.064
Duration of diabetes, years	17.48 ± 11.29	17.34 ± 11.13	17.66 ± 12.19	0.901
HbA1c, %	7.84 ± 1.42	7.73 ± 1.37	8.38 ± 1.53	0.016
Injections/day	2.69 ± 1.27	2.65 ± 1.29	2.90 ± 1.15	0.259
Perception of disinfection importance				
Necessary	211 (88.7)	195 (98.5)	16 (40.0)	0.001
Unnecessary	27 (11.3)	3 (1.5)	24 (60.0)	
Infection symptoms				
Yes	68 (28.6)	60 (30.3)	8 (20.0)	0.250
No	170 (71.4)	138 (69.7)	32 (80.0)	
Employed				
Yes	98 (41.2)	75 (37.9)	23 (57.5)	0.034
No	140 (58.8)	123 (62.1)	17 (42.5)	
Injections outside the home				
Yes	166 (69.7)	134 (67.7)	32 (80.0)	0.135
No	72 (30.3)	64 (32.3)	8 (20.0)	
Type of diabetes				
Type 1	52 (21.8)	41 (20.7)	11 (25.0)	0.636
Type 2	140 (58.8)	118 (59.6)	22 (55.0)	
Unknown	46 (19.3)	41 (20.7)	7 (17.5)	
Needle re-use				
No	221 (92.9)	188 (94.9)	33 (82.5)	0.012
Yes	17 (7.1)	10 (5.1)	7 (17.5)	
Showering or bathing				
Every day	91 (38.2)	76 (38.4)	15 (37.5)	0.815
Every 2–3 days	129 (54.2)	108 (54.5)	21 (52.5)	
Every ≥4 days	18 (7.6)	14 (7.0)	4 (10.0)	

Data are shown as means ± standard deviations or numbers (%). Tests were performed using the χ^2^ test, Fisher’s exact test, or the unpaired *t*-test, as appropriate. HbA1c: glycated hemoglobin.

**Table 2 healthcare-09-00402-t002:** Univariate analyses of clinicodemographic characteristics according to infection symptoms.

Variable	All (*n* = 238) (%)	Infection Symptoms (*n* = 68) (%)	No Infection Symptoms (*n* = 170) (%)	*p*-Value
Sex				
Male	125 (52.5)	28 (41.2)	97 (57.1)	0.031
Female	113 (47.5)	40 (58.8)	73 (42.9)	
Age, years	63.22 ± 12.77	57.22 ± 14.09	65.63 ± 11.39	0.001
Duration of diabetes, years	17.48 ± 11.29	15.02 ± 11.26	18.33 ± 11.19	0.075
HbA1c, %	7.84 ± 1.42	7.93 ± 1.35	7.80 ± 1.45	0.549
Disinfection of injection site				
Yes	203 (85.3)	63 (92.6)	141 (82.9)	0.150
No	35 (17.7)	6 (8.8)	29 (17.0)	
Injections/day	2.69 ± 1.27	3.10 ± 1.22	2.53 ± 1.25	0.001
Employed				
Yes	98 (41.2)	30 (44.1)	68 (40.0)	0.664
No	140 (58.8)	39 (57.4)	102 (60.0)	
Injections performed outside the home				
Yes	166 (69.7)	58 (85.3)	108 (63.5)	0.001
No	72 (30.3)	10 (14.7)	62 (36.5)	
Type of diabetes				
Type 1	52 (21.8)	24 (35.3)	28 (16.5)	0.017
Type 2	140 (58.8)	35 (51.5)	106 (62.4)	
Unknown	46 (19.3)	10 (14.7)	36 (21.2)	
Needle re-use				
No	221 (92.9)	59 (86.8)	162 (95.3)	0.028
Yes	17 (7.1)	9 (13.2)	8 (4.7)	
Showering or bathing				
Every day	91 (38.2)	35 (51.5)	57 (33.5)	0.028
Every 2–3 days	126 (52.9)	28 (41.2)	101 (59.4)	
Every ≥4 days	18 (7.6)	6 (8.8)	12 (7.1)	

Data are shown as mean ± standard deviation or number (%). Tests were performed using the χ^2^ test, Fisher’s exact test, or the unpaired *t*-test, as appropriate. HbA1c: glycated hemoglobin.

**Table 3 healthcare-09-00402-t003:** Multivariable logistic regression analyses of factors associated with infection symptoms.

Variable	Forward Stepwise Selection			Forced Entry		
	OR	95% CI	*p*-Value	OR	95% CI	*p*-Value
Disinfection of injection site	2.552	1.044–6.243	0.040	2.492	0.999–6.216	0.050
Age	0.950	0.928–0.973	0.001	0.955	0.955–0.930	0.001
Injection outside the home	3.055	1.412–6.606	0.005	2.728	1.153–6.455	0.022
Sex	-	-	0.068	-	0.904–3.143	0.100
Type of diabetes	-	-	0.111	-	0.427–1.175	0.182
Injection/day	-	-	0.260	-	0.825–1.471	1.102
Hygiene habits	-	-	0.841	-	0.611–1.825	0.845
Hosmer–Lemeshow test Percentage of correct classifications		*p* = 0.475 73.1%			*p* = 0.269 77.3%	

OR: odds ratio, CI: confidence interval. Only statistically significant variables are presented in Table 3.

## Data Availability

The author has no permission for providing the data currently. In order to provide data to a third-party organization in any way, after acquiring consent from the patients and preparing an experiment plan specifying the provision of providing data to the outside, it is necessary to obtain the approval of the ethics committees; the author cannot make the data publicly available in the prescribed form.

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
