# Peer review of "Factors That Affect Symptoms of Injection Site Infection among Japanese Patients Who Self-Inject Insulin for Diabetes"

_healthcare, 2021, doi:10.3390/healthcare9040402_

Round 1

Reviewer 1 Report

The study investigates the prevalence of infections and the determinants of infections in diabetes patients self-injecting insulin. The major concern is the final conclusion of the authors, which seems to be confusing. All the other and specific remarks are outlined below.

Abstract:

Performing disinfection associated, omitting disinfection associated  - confusing (lines 20-21)

Safe injection technique – not clear where it comes from in the conclusion (line 22)

Introduction

For what kind of intervention (catheters or subcutaneous injections, mentioned in the text before) Wong et al study did not recommend skin disinfection? (line 38)

”there is a large gap between Japan and other countries in attitude and implementation rate of skin disinfection before subcutaneous injection” (lines 61 - 62) requires reference

Methods

“Patients were referred by their physician and were contacted by researchers” (line 84), not clear how this referral occurred. Were the patients selected by the doctors, who then informed the researchers about the time they (researchers) can meet the patients in the hospital?

”We initially identified 307 potential participants, although 69 patients were excluded because of missing data regarding glycated 90 haemoglobin (HbA1c) concentration or infection symptoms at the injection site” (lines 89-91) – not clear how the inclusion happened. Did you have in advance information about the patient symptoms of infection and HbA1c and based on that included patients, or were the patients with lacking information on these variables excluded due missing data in the questionnaire in the analysis stage?

Sample size calculation requires more details (lines 93-94).

The sub-heading 2.3. should be changed  into Questionnaire development. Do you have any information about validity/reliability of your questionnaire? (lines 95-98).

What do you mean “independently associated”? (line 142) Does it mean that you ran adjusted regression models? If so, please stay so.  Also, please provide more information on how the regression modelling was performed.

Results

Generally,  there is some mismatch between the data that were said were collected, and the reported information (e.g. there is said that  the data on where the injection occurred outside home, and  the symptoms of infection, were collected, but there is not information about this in the results. On the other hand, there is reported the days of infection and it is not explained how the data on this variable were collected. Please align.

Table 3 . why do you report B (regression coefficient) for logistic regression? Please report OR for all co-variates in the model.

 Discussion

”The majority of patients (83.2%) performed skin disinfection before the self-injections, which is substantially higher than the rates reported in other countries” (line 193-194) requires reference.

”Thus, patients with poor injection technique may have omitted skin disinfection” (lines 203-203) – why so? How injection technique is related to  skin disinfection?

It is unclear whether these individuals intentionally omitted disinfection based on knowledge of previous studies or results from other countries where omission of disinfection was not associated with skin infection (lines 204-206) – the sentence does not make sense.

Employed patients were also less likely to perform skin disinfection (vs. unemployed patients), although further studies are needed to clarify the relationship (lines 206-208) – quite straightforward explanation, don’t you think?

”Logistic regression analysis revealed that infection at the injection site was significantly associated with skin disinfection before injection, age, and performing injections outside home.” (lines 209-210) – please describe the direction of association: e.g. more often disinfection related do less often/less days of infection, or similar.

“Even taking these results into account, no signs of infection were observed among patients who did not perform skin disinfection. Therefore, omitting skin disinfection before the insulin injection was not associated with infection”(lines 214-215) – direct contradiction to the previous sentence. What do you mean?

"It is possible that skin conditions might be incorrectly interpreted by the patient as an infection symptom, such as bruising that is related to a poor puncture technique [7] (lines 220-221) – move to limitations.

Approximately one-half of our patients who performed injections outside their homes performed the injection in a washroom (lines 227) – this information appears first time in the discussion; adjust the results.  Moreover, why didn’t you do analyses for the risk of infection according to where outside home one has injected her/his insulin?

Limitations. Do you self believe the validity of your results, when you have so many limitations? If yes, why?

Conclusion

"We found that omitting skin disinfection before injection was not associated with infection" (line 253) – why so? You have p=0.040, i.e.  significant disinfection as a co-variate in the model for infections as an outcome (Table 3)?

“A safe injection technique, as well as a suitable environment, are needed for patients with diabetes who must self-inject insulin outside their home” (lines 255-256) – this conclusion is not supported by your results.

Reviewer 2 Report

This observational cross-sectional study evaluated factors that influenced 13 symptoms of injection site infection among 238 Japanese patients. A comparison of patients who did and did not perform skin disinfection revealed significant differences in terms of HbA1c concentration, perception that skin disinfection is necessary, employment status, and needle re-use.      As the authors themselves say,the limitations of the current study are subjectiveness and small sample size.

The data reported in lines 163 and 166 do not correspond to those in Table 1

Round 2

Reviewer 1 Report

I do not see a change in the conclusion in the abstract - still not clear if disinfection is or is not associated with infection.

All the exclusions, since they happened during the analyses process, should be reported in the analysis subsection

Sample size calculation still not clear.

Descriptive table on with/without infection is enough, as infection is your outcome

Please provide more details on how you validated the questionnaire, as well as the results proving its validity

Please argue the choice of your regression modelling (step wise inclusion), please run sensitivity analyses with other kinds of modelling.

Please discus cross-sectional study limitations, e.g. reverse causality.
